# Optimization of the Formulation and Properties of 3D-Printed Complex Egg White Protein Objects

**DOI:** 10.3390/foods9020164

**Published:** 2020-02-08

**Authors:** Lili Liu, Xiaopan Yang, Bhesh Bhandari, Yuanyuan Meng, Sangeeta Prakash

**Affiliations:** 1College of Food and Bioengineering, National Experimental Teaching Demonstration Center for Food Processing and Security, Henan University of Science and Technology, Luoyang 471023, China; yangliuyilang@126.com (L.L.); yxp58166@163.com (X.Y.); zfcyuan0901@163.com (Y.M.); 2School of Agriculture and Food Sciences, The University of Queensland, QLD 4072, Australia; b.bhandari@uq.edu.au

**Keywords:** egg white protein, 3D printing formulation, rheological properties, sensory evaluation score, viscosity

## Abstract

The 3D printing of foods is an emerging technique for producing unique and complex food items. This study presents the optimization of a new formulation for 3D printing foods on the basis of a complex system, which contains egg white protein (EWP), gelatin, cornstarch, and sucrose. The effects of different formulations on the rheological properties and the microstructure of the printing system were investigated. The formulation was optimized through response surface methodology, and a central composite design was adopted. The optimum formulation of the 3D mixture printing system was made of gelatin (14.27 g), cornstarch (19.72 g), sucrose (8.02 g), and EWP (12.98 g) in 250 mL of total deionized water with a maximum sensory evaluation score of 34.47 ± 1.02 and a viscosity of 1.374 ± 0.015 Pa·s. Results showed that the viscosity of the formulation correlated with the sensory evaluation score. The rheological properties and tribological behavior of the optimum formulation significantly differed from those of other formulations. A viscosity of 1.374 Pa·s supported the timely flow out of the printing material from the nozzle assisting 3D printability. Thus, 3D printing based on the egg white protein mixture system is a promising method for producing complex-shaped food objects.

## 1. Introduction

Customized foods are increasingly becoming popular as consumers are demanding healthy, tasty, and great-looking food [1]. Two options are available to create completely customized foods: using a material set that is large enough to satisfy all consumers’ wants or using a small material set that can be combined at varying ratios. Therefore, food printing is a method to distribute food in a personalized manner and satisfy this demand. 3D printing technology, also known as additive manufacturing or rapid prototyping, is an emerging and promising technique to create an object through layer-by-layer fabrication [2]. Many manufacturing applications involving this technology have been used in different fields, such as automobile, aircraft, and medical services [3,4]. This technology also shows potential for applications involving the production of food products with good quality control, environmentally friendly, high energy efficiency, and low cost. Suitable materials can be mixed using 3D printing technology and then processed into various intricate shapes and structures, which are impossible to be made or uneconomical to produce under traditional manufacturing practices [5]. With 3D printing technology, food processing or preparation technologies have emerged in the digital age [6].

Several studies have investigated food printing. Most of them have focused on fabricating novel food items [7]. Some researchers explored fundamental-level issues in food printing, such as converting ingredients into tasty products for healthy and environmental reasons [8]. Thus, the properties and composition of food materials during 3D printing should be understood. These materials must be flowable to extrude from a nozzle and hold their structure to maintain shape during and after printing [9]. Applying multiple materials is quite common in food design and fabrication, and the diversity of printing materials empowers consumers to take control of food design. Bhandari and Roos [10] reported that plasticization decreases the glass transition temperature of food polymers, such as starch, proteins, and carbohydrates; consequently, they become viscous printable materials. Lam et al. [11] also developed a unique blend of starch-based polymer powders (i.e., cornstarch, dextran, and gelatin) for specific 3D printing. Purees, gels, and dough are deposited directly without adding a structuring agent or hydrocolloids as deposited materials to support their structures [12]. Some researchers also mixed gelling agents, such as carrageenan, xanthan gum, and gum arabic, with other ingredients, including supplements, and then mix them to create edible constructs, such as raspberry domes, mushroom-shaped bananas, and milk cubes [13,14].

Egg white protein (EWP) is a food ingredient with multiple functional properties, such as foaming, emulsification, heat-set gelation, and binding adhesion [15,16]. Therefore, EWP is used as a functional ingredient in numerous food products, such as meringues, mousses, and bakery products [17,18,19,20]. EWP is a promising food material for developing various 3D constructs because it can form a heat-induced edible gel [21], which is conducive to further thermal processing. Thus, in this study, the optimization of an EWP printing mixture system was investigated.

Changing the recipe of a food item can considerably affect the printability of a mixture system and the shape stability of final products. Therefore, optimizing the 3D printing process and exploring new food materials or recipes suitable for printing are potential trends and challenges in the field of 3D food printing. A printable material requires rheological properties that allow its extrusion through a small needle and rapid stabilization after deposition to guarantee the fidelity of the shape of the extruded line [22]. These requirements imply that hydrogel materials with shear-thinning properties and robust yield-stress behavior are attractive candidate materials for 3D food printing. Food tribological studies have focused on the relation between tribological and sensory attributes. They can provide further insights into fundamental food behavior, particularly during oral processing. 

Although numerous reports about 3D printing have been provided, the optimum formulation based on EWP has yet to be published. In our previous research, we learned the functional properties of EWP and its influence on 3D mixture printing system [23,24,25]. This present work aimed to explore a new formulation of a mixture system for 3D printing. Various materials, such as gelatin, cornstarch, and sucrose, were added to the mixture to create printing mixture systems with a certain viscosity, thereby improving the suitability of EWP for 3D printing. The mixture formulation was optimized through response surface methodology (RSM). The formulation parameters were EWP, gelatin, cornstarch, and sucrose. Sensory score was used as an index. This study also aimed to analyze the relationship between the sensory evaluation score and viscosity of the printing system. The rheological and tribological behaviors of the optimized formulation were compared with those of other formulations. 

## 2. Materials and Methods

### 2.1. Materials

Egg albumen protein powder with 80% protein concentration was obtained from All Food Systems Australia Pty. Ltd. (Queensland, Brisbane, Australia). Edible bovine gelatin (bloom strength 250) was purchased from GELITA Australia Pty. Ltd. (Queensland, Brisbane, Australia). Cornstarch and sucrose were purchased from Coles Supermarket (Queensland, Brisbane, Australia). 

### 2.2. 3D Printing

In this study, the 3D printing process of the mixture was investigated using the 3D printer instruction (Kunshan PORIMY 3D Printing Technology Co., Ltd. Jiangsu Province, China). The 3D printing system was composed of three major parts: (i) a mechanical platform, (ii) an electrical and software component, and (iii) an extruder and cooling system. Figure 1 shows the nozzle size and the complete 3D printer with the electronic feed hopper, the extruder, and the cooling systems.

The following printing parameters were left unchanged throughout the tests to ensure the ability to print complex 3D objects: nozzle diameter size of 1.0 mm, moving speed of 70 mm/s, extrusion rate of 0.004 cm^3^/s, nozzle height of 3.0 mm, printing at 40 °C, and no forced air flow. 

### 2.3. Experimental Design and Optimization of the Printing Formulation 

The effect of changing a single factor on sensory evaluation was initially analyzed to determine the preliminary range of the added amount of the variables. The added amount of the formulation material was set as follows: gelatin (10 g), cornstarch (20 g), sucrose (10 g), EWP (15 g), and total deionized water (250 mL). The variable parameters were as follows: gelatin (i.e., 7, 9, 11, 13, 15, and 17 g), cornstarch (i.e., 14, 16, 18, 20, 22, and 24 g), sucrose (i.e., 6, 8, 10, 12, 14, and 16 g), and EWP (i.e., 9, 11, 13, 15, 17, and 19 g). Then, the formulation parameters were optimized through RSM. A central composite design (CCD) with four independent variables at five levels was prepared to determine the combined effect of the independent variables on the response. The range and the levels of the variables investigated in this study are given in Table A1. The independent variables and their levels were chosen on the basis of the results of preliminary experiments. Viscosity and sensory evaluation were the dependent variables. A total of 36 experimental runs (i.e., 16 factorial points, 8 axial points, and 12 center points) were generated on the basis of single-factor experiments by using a CCD five-level-four-factor design for optimizing the formula conditions as shown in Table A2. A total of 12 replicates (treatments 25–36) at the center of the design were used to estimate the pure error sum of squares. Data from the CCD were explained via multiple regressions to fit the following second-order polynomial equation:
(1)$$Y=\beta_{0}+\sum_{i=1}^{4}\beta_{i}X_{i}+\sum_{i=1}^{4}\beta_{ii}X_{i}^{2}+\sum_{i=4}^{3}\sum_{j=i+1}^{4}\beta_{ij}X_{i}X_{j},$$
where *Y* is the dependent variable (sensory score); *β*_0_ is a constant; *β*_i_, *β*_ii_, and *β*_ij_ are regression coefficients, and *X*_i_ and *X*_j_ are the levels of the independent variables. ANOVA was estimated with Design-Expert 8.05b (trial version, State-Ease Inc., Minneapolis, USA). The degree of fitting and significance test of regression equation combined with RSA were calculated, and a regression model was established [26,27]. In accordance with the response surface methodology and a contour map, the influence of each factor and interaction function to sensory evaluation was analyzed, and the optimized 3D printing formulation was confirmed. *p* < 0.01 indicated statistically significant differences. Data were determined in triplicate, and results were averaged.

### 2.4. Preparation of the Complex Printing System

The components of the material formula for 3D printing included different concentrations of bovine gelatin, cornstarch, sucrose, and EWP with 250 mL of deionized water. The composition formulations of the 3D printing EWP mixture systems are given in Table A1. 

EWP powders were dissolved in 50.0 mL of deionized water in a water bath (55 °C, 5 min) and stirred at 7000–8000 rpm by using a T-25 digital Ultra-Turrax homogenizer (IKA, Germany) to prepare the above different concentrations (pH 6.5). Bovine gelatin, cornstarch, and sucrose powders were dissolved with deionized water in a water bath (80 °C, 10 min), stirred at 450 rpm through a mixer (JJ-1A digital Electric Mixer, Changzhou Instrument Factory, China), and subsequently mixed and heated in a water bath (100 °C, 20 min, and 300 rpm). The different mixture systems were cooled at 55 °C for 5 min. Then, the above EWP solutions were added with stirring (55 °C, 10 min, 400 rpm) to prepare the printing complex mixture systems. The temperature of the printing systems was kept at 40 °C for 20 min before printing. These printing mixture systems were used for all analyses.

### 2.5. Sensory Evaluation

The 3D-printed objects heated for 3 min in a microwave oven (Panasonic NNDS592BQPQ, Microwave Oven) were subjected to sensory evaluation by 30 trained panelists (15 females and 15 males, 25–30 years old, healthy, and lactose tolerant). Before sensory evaluation was conducted, the panels were trained by using related 3D-printed products to familiarize them with the rating method, terminology for each attribute, and sensory characteristics. The samples were coded with three digits, and the panelists were instructed to evaluate the appearance, flavor, taste, texture, and overall acceptability score by using a seven-point hedonic scale ranging from “1 = extremely dislike” to “7 = extremely like” in accordance with a previously described method [28,29]. Based on the same parameter weight, the total sensory score was equal to the sum of the five scores. The assessors randomly evaluated the coded 3D-printed samples. Before tasting each sample, they cleansed their palate with cold, filtered tap water. Products were characterized under daylight illumination and in isolated booths within a sensory laboratory.

### 2.6. Property Analysis of Different Formulations 

#### 2.6.1. Relationship between Viscosity and Sensory Evaluation

The viscosities of all the formulations were tested using an AR-G2 rheometer (AR-1000, Co. TA, USA) with a stainless-steel plate (60 mm in diameter). For a wide range of food products from Newtonian fluid to thick emulsion, a close relationship exists between viscosity and sensory perception at a shear rate of 50 s^−1^ [30]. Considering that the appearance, texture, and overall acceptability score of 3D-printed objects were mainly affected by viscosities, the sensory score was selected as the sum of the three scores. Table A2 and Table A3 show the viscosities at a shear rate of 50 s^−1^ (η_50_) and the sensory scores for all the 3D printing systems. 

#### 2.6.2. Rheological Measurement 

The optimum formulations (Test OF, at a viscosity of 1.374 Pa·s), Test 1 (at the lowest viscosity of 0.483 Pa·s), and Test 24 (at the highest viscosity of 2.058 Pa·s) were prepared as previously described (40 °C). All rheological measurements were performed using an AR-G2 rheometer (AR-1000, Co., TA, USA) with a stainless-steel plate (60 mm in diameter). The gap between the two plates was set to 1000 μm. The shear rate was ramped from 0.1 to 500 s^−1^ at 40 °C for 10 min to determine the flow behavior. Temperature sweep tests were performed from 30 to 95 °C at a scan rate of 5 °C/min and a shear rate of 50 s^−1^. Dynamic viscoelastic properties were characterized at the small amplitude oscillatory frequency sweep mode. The frequency was oscillated from 0.1 to 100 rad/s at 40 °C, and measurements were performed within the identified linear viscoelastic region and at a 0.5% strain [31]. Elastic modulus (G′) and loss modulus (G″) were recorded. Data were reported as the average of three replicates.

#### 2.6.3. Tribological Measurement

The three different formulations (i.e., Test OF, Test 1, and Test 24) were subjected to tribological measurements on a Discovery Hybrid Rheometer by using a ring on a plate tribo-rheometry (TA Instrument, USA) on a rough plastic surface of 3M Transpore Surgical Tape 1527-2 (3M Health Care, USA). The hydrophobic rough surface of human tongue was modeled using 3M Transpore Surgical Tape 1527-2 [32]. The tape was cut in a square shape. It was then placed and pressed firmly on top of the low plate geometry. After each measurement, the tape was replaced, and the probe was cleaned with deionized water and dried with laboratory wipes. 

The temperature sweep tests were conducted from 20 to 90 °C at a scan rate of 5 °C/min and a shear rate of 5 rad/s. The tribological measurements were performed at 40 °C for 10 min. The in-mouth force was between 0.01 and 10 N [23], so we used normal forces of 2 N to represent the moderately normal force applied to the samples during oral processing. The samples were pre-sheared at a speed of 0.01 rad/s for 1 min and equilibrated for 2 min before each measurement was performed. The results were recorded at rotational speeds of 0.01 to 30 rad/s with 10 points per decade, and the resulting friction coefficients were recorded. Three replicates of each sample were prepared. 

### 2.7. Scanning Electron Microscopy (SEM) of the Printing Mixture Systems

The morphological characteristics of the printing mixture systems were observed using an Ultra Plus scanning electron microscope (Zeiss, Oberkochen, Germany). The dried printing mixture samples were mounted on a conducting resin with 2–3 mm thickness [33]. The magnification of the observations was ×800 by taking three different images.

### 2.8. Statistics

Differences among groups were analyzed through one-way ANOVA with Tukey’s honestly significant difference by SPSS software 22.0 (SPSS, Chicago, IL, USA). *p* < 0.05 was statistically significant.

## 3. Results and Discussion

### 3.1. Optimization of 3D Printing Formulation 

#### 3.1.1. Single-Factor Results

The effects of different added amounts of formula materials on the sensory evaluation of 3D printing products are shown in Figure 2. The sensory score increased significantly as the amount of added gelatin and cornstarch increased and peaked at 13 and 18 g, respectively. Thereafter, the score did not significantly decrease (Figure 2A,B). Therefore, 13 and 18 g were selected as the amounts of gelatin and cornstarch added for further experiments, respectively. Figure 2C shows that the sensory score increased as the amount of sucrose added increased between 6 and 8 g. Afterwards, the score gradually decreased as the added amounts increased. Therefore, 8 g of sucrose was sufficient to obtain the maximum sensory score. Figure 2D illustrates that the sensory evaluation of the 3D-printed products significantly increased from 21.15 to 33.40 when the amount of the added EWP varied from 9 to 13 g and slightly increased when the added amount exceeded 13 g. Although the score was also high at 15 g, increasing the amount of added EWP also increased the cost industrial material process. Thus, 13–15 g was the amount deemed favorable for material formulation. In accordance with the single-parameter study, the addition amounts of 11–15 g gelatin, 16–20 g cornstarch, 6–10 g sucrose, and 11–15 g EWP were adopted for the RSM experiments.

#### 3.1.2. Analysis of Response-Surface Design 

Based on the single factor test, the design matrix and the corresponding results of RSM experiments are shown in Table A1, Table A2 and Table A3. Through multiple regression analysis on the experimental data, the model for the predicted response was obtained by the quadratic polynomial equations. The ANOVA (Table A4) results of the quadratic model suggest that the model was highly significant and reliable. The detailed analysis of this section see the Appendix A.

### 3.2. Effects of the Amounts of the Added Materials on the Sensory Evaluation 

The interactions among different independent variables and their corresponding effects on the response, response surface (3D), and contour plots were drawn (Figure 3). Response surface methodology was performed to illustrate the effects of the amounts of added gelatin, cornstarch, sucrose, and EWP on the response. The regression model equation was used to predict of the effects of the four parameters on the sensory score. The types of interactions between the four tested variables and the relationship between responses and experiment levels of each variable are illustrated in the 3D response-surface plots of the response surfaces [34]. A contour plot is a graphical representation of a 3D response surface as a function of two independent variables, maintaining all other variables at a fixed level [35]. These plots can help understand the main and interaction effects of the independent variables on the response.

Figure 3A shows the 3D graphic surface and contour plot of the combined effects of gelatin (*X*_1_) and sucrose (*X*_3_) on the sensory score. These plots present the response as a function of two factors, keeping the other variable constant at its middle level (center value of the testing ranges). The tortuose surface and the oval contour plot show a strong interaction between these two factors. The sensory score initially increased by increasing the added amounts of gelatin and sucrose but subsequently decreased. This result demonstrated that the effect of gelation (*X*_1_) and sucrose (*X*_3_) on sensory score was significant and in good agreement with the results in Table A3. The sensory score increased as the amount of gelatin increased from 13 to 15 g and sucrose from 7 to 9 g, respectively.

Figure 3B shows the interactive effects of gelatin (*X*_1_) and EWP (*X*_4_) on the sensory score. The maximum predicted value indicated by the surface was confined in the smallest ellipse in the contour diagram. The smallest ellipse in the contour plot indicated the perfect interaction between the independent variables. The sensory score increased readily as the amount of EWP increased up to 12 g and slightly decreased at high addition amounts. The sensory score was increased by increasing the amount of added gelatin. This phenomenon was most likely due to improvement in viscosity when high amounts of gelatin were added, and gelatinization increased the viscosity of the 3D printing system. A quadratic effect was observed in the added amounts of gelatin and EWP. In Figure 3B, the maximum amounts of gelatin and EWP added were approximately 15 and 15 g, respectively. A high sensory score was obtained when the added amounts of EWP were between 12 and 14 g and when the added amounts of gelatin were between 14 and 15 g.

In a multicomponent system, the change in proteins, carbohydrates, and water affects the melting behavior and plasticization of the food materials during 3D printing [36]. Plasticization decreases the glass transition temperature of food polymers, such as starch, proteins, and carbohydrates; as such, they become viscous printable materials. The maximum predicted response value for the sensory score was 34.45, which was attained under the following optimal formulation conditions: 14.27 g of gelatin, 19.72 g of cornstarch, 8.02 g of sucrose, and 12.98 g of EWP. The model was validated by performing the experiment three times under the optimal conditions. The mean of the sensory evaluation score was 34.47 ± 1.02, and viscosity was 1.374 ± 0.015 Pa·s. A good agreement between the predicted value and the experimental value validated the reliability of the RSM technique for optimizing the process.

### 3.3. Relationship between Viscosity and Sensory Evaluation

The relationship between rheological properties and sensory attributes for the 36 test samples of CCD is shown in Table A2 and Table A3. For many food products from Newtonian fluid to thick emulsion, the viscosity at an oscillatory frequency of 50 s^−1^ is closely related to sensory [37]. A good correlation with complex viscosity is obtained from oscillatory small deformation experiments at a frequency of 50 rad/s [38]. Kravchuk et al. [39] found that food is exposed to a range of shear and deformation processes during oral processing, which cause its different responses to flow behavior. In our study, the viscosity of the rheological parameter demonstrated a certain correlation with a sensory score at 50 s^−1^ or 50 rad/s. In Figure 4, the correlation coefficient of viscosity and sensory evaluation was 15.2012 (*R*^2^ was 0.9457) when the viscosity increased from 0.483 to 1.375 Pa·s, which showed a significantly positive correlation between the two indices. However, when the viscosity was higher than 1.375 Pa·s, viscosity and sensory evaluation had a negative correlation with a correlation coefficient of −10.1914 (*R*^2^ was 0.9509). This result suggested that viscosity was an important parameter in 3D printing.

Figure 5 shows the different geometrical shapes of 3D printing mixture formulations with different viscosities. These results indicated that 3D printing systems with low or high viscosities were unsuitable for printing. During 3D printing, the properties and compositions of food materials are considered the most important factors [36]. Godoi et al. [3] reported that these materials should be homogeneous with appropriate flow properties for extrusion and could support their structure during and after printing.

### 3.4. Rheological Behavior 

Rheology is a study of the deformation and flow behavior of materials. Rheology is especially concerned with how viscosity and viscoelastic behavior change with applied stress or strain. Thus, the rheological property of 3D printing formulations is an important determinant to support the 3D network structure.

The apparent viscosity curves of the three formulations are shown in Figure 6A. Viscosity significantly decreased as the shear rate increased. This result indicated that the three formulations were pseudoplastic fluids with shear-thinning properties. The high viscosity could inhibit the fluidity of the mixture. Therefore, the Test OF would be helpful for the 3D printing system to flow through the nozzle and obtain viscous post deposition for maintaining its shape. In Figure 6B, the viscosities of three test samples started to increase significantly at approximately 75 °C and then decreased sharply at temperatures higher than 85 °C. This coagulation temperature range involved egg white proteins, which were denatured and partially aggregated at the temperature range [40]. Thus, the mixture printing system had a gelling property similar to that of protein, which is heat-set gelation [41]. Furthermore, G′ was higher than G″ in the linear viscoelastic region (Figure 6C,D), suggesting the mixture systems could form elastic gel or a gel-like structures. In addition, G′ and G″ progressively increased as the increasing oscillatory frequency, leading to the increased internal friction of the material. G′ and G″ continuously increased as the viscosity of the different formulations increased at any oscillatory frequency. This might be due to the addition of different formulations materials that generated additional inter- and intra-molecular forces in the mixture system. An ideal printing system should maintain the shape of the extrudate, be printable below the maximum extrusion pressure of the printer, and capable of fusing with earlier printed layers [32]. The increased G′ and G″ of the mixture systems with increasing viscosities of test samples might hinder the printing mixture to flow out from the nozzle. 

### 3.5. Tribological Behavior

The application of food tribology to predict the mouth feel properties of foods has been widely considered by many food scientists. Although this study was not about discovering the link between tribology and any particular sensory characteristic of 3D printing systems, tribological studies have presented other data that can be beneficial to understanding the fundamental food behavior, particularly in the area of oral processing. The tribological behavior of solutions can be regarded as a friction curve, namely, the Stribeck curve, which traditionally consists of three regimes: boundary, mixed, and hydrodynamic. This curve presents relationships between the coefficient of friction (CoF) and the sliding speed. However, for the 3D printing mixture tested in this study, the internal structure was complex with the gelatinization of starch and cross-linked gel network formed by denatured gelatin and egg white proteins. 

The average friction curves for all of the samples at 2 N and 40 °C are shown in Figure 7A. The friction curves of printing mixtures with three test samples of different formulations showed a “stick and slide” pattern. The friction curves did not resemble the traditional Stribeck curve. At low sliding speeds (100–500 μm/s), the mixture system flowed through the gap between two surfaces, but no samples could attach to the surfaces to build a lubrication firm because of the low speed, thereby causing a negligible increase in CoF. When the sliding speed reached 500 μm/s, the CoF decreased as the speed increased because the mixture system that can come into a contact zone to partly separates two surfaces [32,42]. The friction behavior in this regime is primarily affected by interactions between sliding surfaces, although the lubricant plays a role in friction behavior. By contrast, in the hydrodynamic regime observed at high sliding speeds, the lubricant completely separates the sliding surfaces in which a viscous drag of the lubricant dominated the friction response. The mixed regime marks the transition between the boundary and hydrodynamic regimes; the friction coefficient decreases as the sliding surfaces are separated by the lubricant. However, at a high speed (>100,000 μm/s), the CoF increased because it entered the hydrodynamic regimes. The increased speed resulted in the samples being drawn into the contact zone to partly separate the two surfaces [42]. The samples with a high viscosity have a low CoF [32]. However, in our study, Test 24 with the highest viscosity (Figure 7A) had the highest CoF. 

Figure 7B shows that CoF gradually increased as the temperature increased from 20 to 60 °C and began to decrease when the temperature reached 60 °C. This phenomenon might be attributed to the mixture systems that reached the gel-like states as temperature increased and turned semisolid at approximately 60 °C. However, the mixture systems returned to the gel forms when the temperature reached 70 °C, and the samples between the two surfaces were squeezed out, causing CoF to decrease. The gelation of EWP in the mixture systems might be responsible for this change. In our previous study, it was shown that the addition of EWP could significantly increase the CoF of the 3D printing system [23]. Studying these changes in the tribological print properties is conducive to optimizing the printing formulation and helpful for understanding the taste of the 3D-printed food. 

### 3.6. SEM Image of the Printing Systems

The mechanism that underlies the formulation-induced changes in the material properties of the printing system was analyzed by taking microscopic images of printing mixture systems with different viscosities. Figure 8 shows that the material molecules had heterogeneous distribution throughout the stabilized printing systems, which showed flocculation signs. The microstructures of the three formulations showed some differences. When different raw materials such as gelatin, EWP, and cornstarch were added into the three formulations, these raw materials were mixed together to form the complex 3D printing systems. During printing process, cold-induced gels of gelatin were mainly benefit for shaping of printing product. EWP could form heat-induced gels and cornstarch gelatinized, which had excellent shape-retention properties that stabilized the shapes during the late stages of post-printing heating. As the internal structure of 3D-printed objects was complex with the gelatinization of starch and cross-linked gel network formed by denatured gelatin and egg white proteins, SEM showed that there was no significant difference among the microstructures of three formulations. 

## 4. Conclusions 

As an emerging field in additive manufacturing, 3D food printing will continue to gain momentum because it is evolving from a niche high-end novelty market to a practical method for individuals with specific food requirements. This study showed that RSM was a successful technique for optimizing 3D printing formulation by using a melt extrusion 3D printer. This printing system was mainly based on the gelatinization of starch, the characteristics of gelatin cooling solidification, and EWP with thermal denaturation to maintain complex 3D geometries. The response surface and contour plots in RSM were effective in estimating the effects of four independent variables (i.e., amounts of added gelatin, cornstarch, sucrose, and EWP). The four materials significantly affected the viscosities and sensory scores of 3D printing systems. The viscosities increased with the increase of their addition amounts, but the sensory scores increased at first and then decreased. With a total water content of 250 mL, the optimum values were 14.27 g of gelatin, 19.72 g of cornstarch, 8.02 g of sucrose, and 12.98 g of EWP with a sensory evaluation score of 34.47 ± 1.02. The experimental value agreed satisfactorily with the model predicted value. This study aimed to comparatively examine the influence of three formulations with different viscosities on the rheological and frictional properties of the 3D printing system containing cornstarch, gelatin, sucrose, and EWP. Our results showed that adding different materials could significantly change the rheological and frictional behavior of the printing system. Using this design, we successfully printed complex 3D geometric shapes. However, because of the complexity of the printing systems caused by the different amount of the raw materials, the analysis of how the four materials affect the 3D printing system needs to be further studied.

## Figures and Tables

**Figure 1 foods-09-00164-f001:**
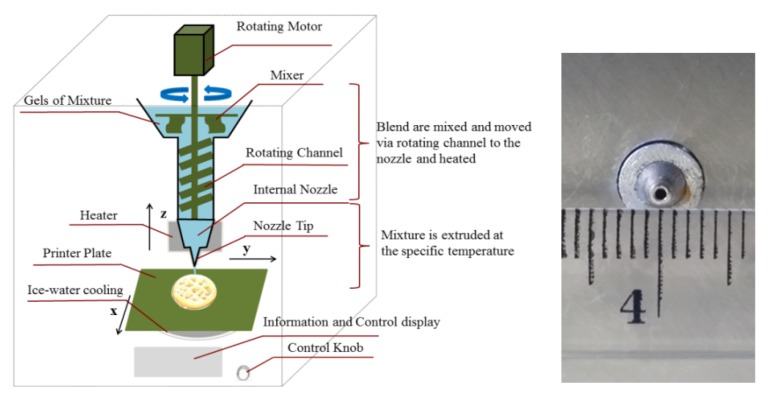
Schematic and nozzle size of the PORIMY 3D printer. The outline dimension is 380 × 390 × 610 mm (width × length × height).

**Figure 2 foods-09-00164-f002:**
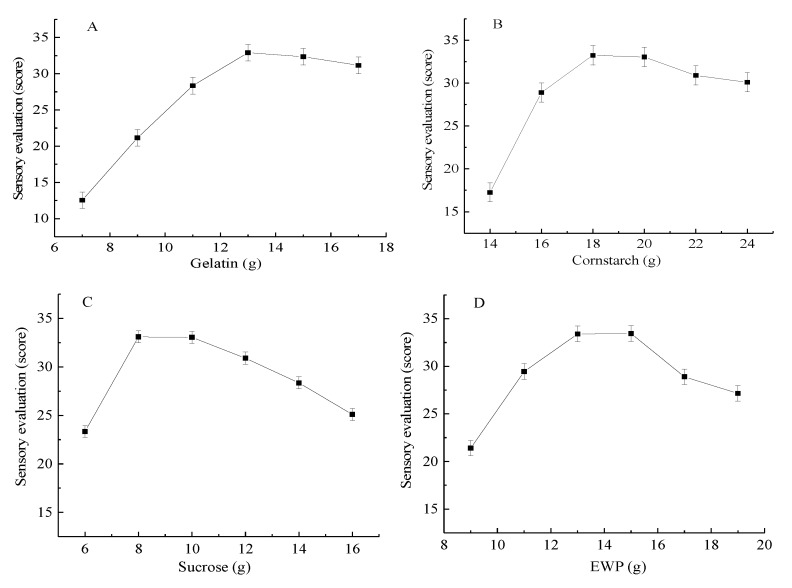
Effects of different amounts of added (**A**) gelatin, (**B**) cornstarch, (**C**) sucrose, and (**D**) egg white protein (EWP) on the sensory evaluation of 3D printing products.

**Figure 3 foods-09-00164-f003:**
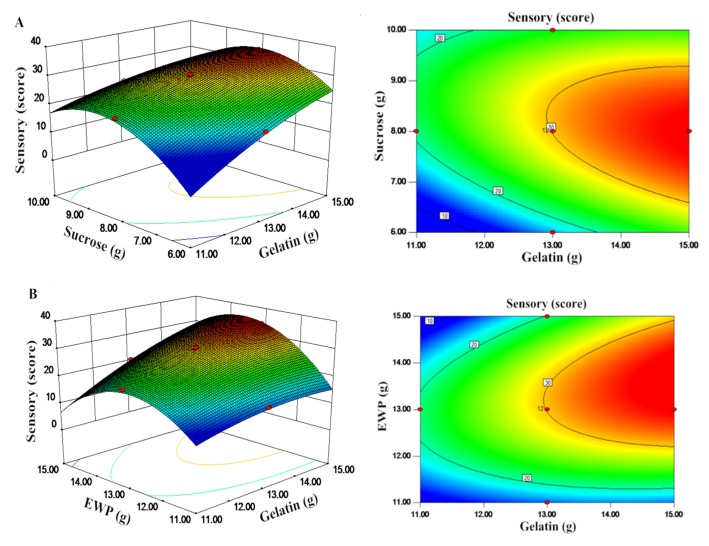
(**A**) Interactive effects of the amounts of the added gelatin and sucrose on the sensory score; (**B**) the interactive effects of the amount of the added gelatin and EWP on the sensory score.

**Figure 4 foods-09-00164-f004:**
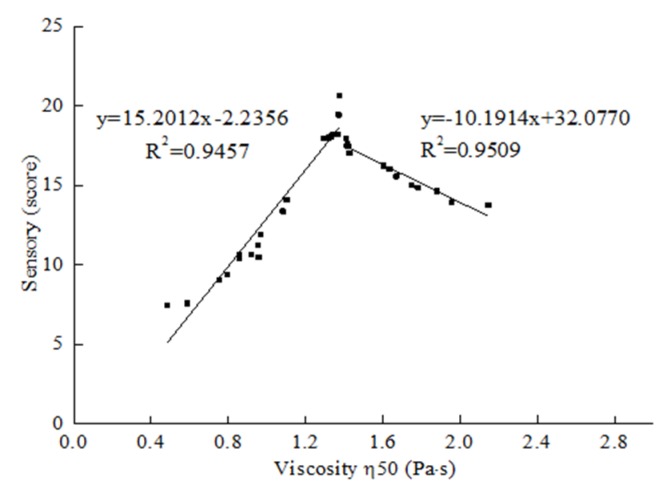
Relationship between sensory evaluation and viscosity 50 (Pa·s). Sensory score was the sum of appearance, texture, and overall acceptability score (Table A3).

**Figure 5 foods-09-00164-f005:**
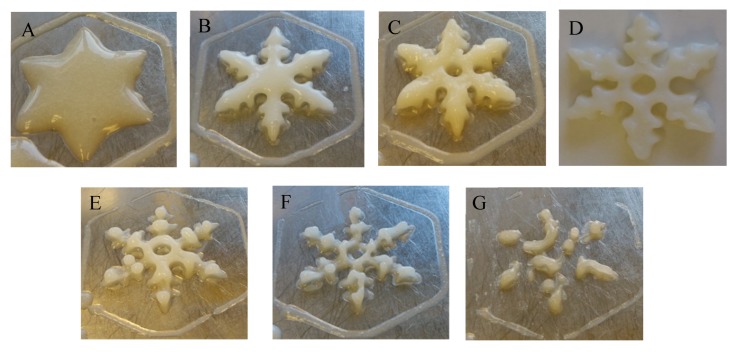
Different geometrical shapes of the selected 3D printing formulation samples with different viscosities. (**A**) 0.483 Pa·s (Test 1), (**B**) 0.587 Pa·s (Test 2), (**C**) 0.753 Pa·s (Test 19), (**D**) 1.374 Pa·s (Test OF, the optimum formulation), (**E**) 1.814 Pa·s (Test 22), (**F**) 1.885 Pa·s (Test 11), (**G**) 2.058 Pa·s (Test 24). The extrusion parameters were as follows: nozzle diameter of 1.0 mm, nozzle height of 3.0 mm, nozzle moving speed of 70 mm/s, and extrusion rate of 0.004 cm^3^/s.

**Figure 6 foods-09-00164-f006:**
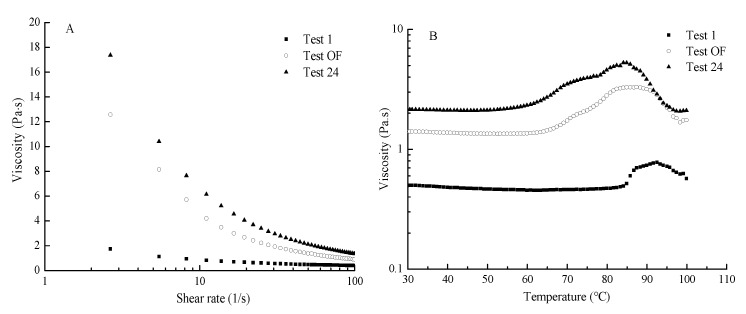
Rheological behaviors. (**A**) Apparent viscosity, (**B**) effect of temperature on viscosity, (**C**) G′, and (**D**) G″) of the different printing systems (Test OF, Tests 1 and 24 at the viscosity of 1.374, 0.483, and 2.058 Pa·s).

**Figure 7 foods-09-00164-f007:**
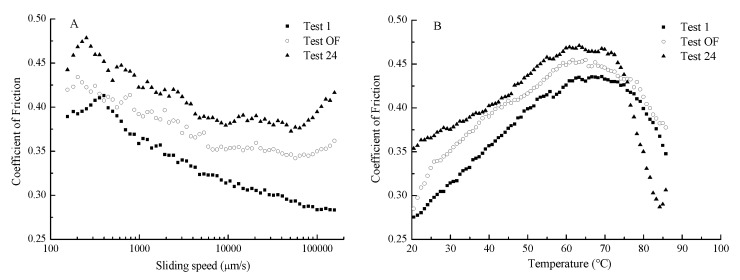
Frictional properties (**A**) Apparent friction, (**B**) Effect of temperature on friction) of the different printing systems (the optimum formulation, Tests 1 and 24 at the viscosity of 1.374, 0.483, and 2.058 Pa·s).

**Figure 8 foods-09-00164-f008:**
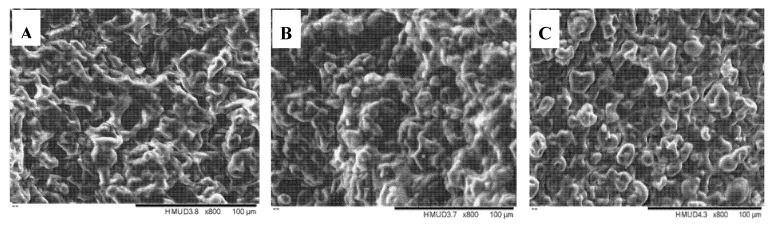
SEM image of the mixture systems (×800). (**A**) Test OF at a viscosity of 1.374 Pa·s, (**B**) Test 1 at a viscosity of 0.483 Pa·s, and (**C**) Test 24 at a viscosity of 2.058 Pa·s.

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
