# Peer review of "Optimization of the Formulation and Properties of 3D-Printed Complex Egg White Protein Objects"

_foods, 2020, doi:10.3390/foods9020164_

Round 1
Reviewer 1 Report
Recommendation
Major revision
I have reviewed this manuscript
Optimization of the formulation and properties of
3D-printedcomplex egg white protein objects
Lili Liu, Xiaopan Yang, Bhesh Bhandari, Yuanyuan Meng and Sangeeta Prakash
The authors have prepared several formulations out of egg white protein (EWP), gelatin, cornstarch, and sucrose and have conducted sensory, rheology and tribology analyses after 3D printing them. Inevitably when reviewing a manuscript the tendency is to focus on the negatives however I want to emphasize strongly that in this case there are a large number of very positive aspects including excellent structure, presenting the methodology in detail, interpreting the results in depth.
Major Issues
It seems that for the sensory analysis the 3D-printed objects were presented to the panel. Why? It would be very interesting if a non-printed object (for example moulded) would also be given to the panel and a brief comparison between the printed and non-printed samples had also been presented in the paper.I suggest the section “Analysis of response-surface design” is moved to an appendix and only one or two sentences are brought in the main text. It is too iterative (repetitive) and distracting (diverting attention) in the middle of the manuscript Line 243, I think the following line should appear somewhere earlier in the paper
“ANOVA is a statistical technique that splits the total variation in a set of data into component parts related to particular sources of variation to test the hypotheses on the parameters of the model[24].”
Line 371-374, these sentences are not clear and if I understood it well, the conclusion is not correct either“G′ and G″ continuously and significantly increased as the viscosity of the different formulations increased at any oscillatory frequency. This increase might be due to the different materials of formulations in the mixture that caused an increase in the number of inter- and intra-molecular forces.”
Line 412-419, it is not clear. I suggest it be revisited.
“This phenomenon might be attributed to the mixture system that reached a gel-like state as temperature increased turned semisolid at approximately 60 °C. However, the mixture system returned to a gel form when the temperature reached 70 °C, and the samples between the two surfaces were squeezed out, causing CoF to decrease. At the cooling stage, CoF was stable between 40 °C and 20 °C and increased when the temperature was lower than 20 °C. The gelation of gelatin in the mixture system might be responsible for this change. Studying the changes in the tribological print properties is conducive to optimizing the printing formulation and helpful for understanding the taste of the 3D-printed food.”
Line 429-431, it is not clear from the SEM pictures. I could not see any distinctive difference between the microstructures.
“The SEM of the printing system with a high viscosity showed a compact microstructure, indicating that EWP could form heat-induced gels and had excellent shape-retention properties that stabilized the shape of 3D-printed products during the late stages of post-printing heating.”
Moreover, I ask that the authors specifically to revisit the following lines.
Minor Issues
Title: “3D-printedcomplex egg” should be changed to “3D-printed complex egg”
Line 104-105, Figure 1. There are some texts on the Figure that cannot be read (94, T02?)
Line 33-“food” looks redundant. It has been brought again in the next line.
Line 44- “flow able” should be changed to “flowable”
Line 85-“2.2.3. D printing” should be changed to “2.2 3D printing”
Line 86-87- This sentence should be revisited “In this study, the 3D printing of the mixtures was investigated in accordance with the chocolate 3D printer instruction (Kunshan PORIMY 3D Printing Technology Co., Ltd. Jiangsu Province, China).”
Line 121-“viamultiple” should be changed to “via multiple”
Line 146-“NNDS592BQPQPanasonic” should be changed to “Panasonic NN-DS596BBPQ”
Line 147-“30trained panelists" should be changed to “30 trained panellists”
Line 219-“the addition amounts11–15 g of gelatin” should be changed to “the addition amounts of 11–15 g gelatin”
Line 246-“3Dprintedproducts” should be changed to “3D printed products”
Line 251-“0.9950impliedthat” should be changed to “0.9950 implied that”
Line 253-255 and Line 260-266, Line 289 the text font is different from the other parts
Line 369-“ Figs. 6C and D),” should be changed to “(Figs. 6C and D),”
Line 422 and 423, Figure caption, “aviscosity” should be changed to “a viscosity”.
Line 442 “scoreof6.87±0.03” should be changed to “score of 6.87±0.03”
Reviewer 2 Report
This is an interesting study in a very active emerging area of food science (3D printing of food), but the main lack of this paper is the use that the authors made of the concept of sensory evaluation (score). To evaluate the appearance, flavor, texture, taste, and overall acceptability score the authors use a seven-point hedonic scale but after they don’t define how they obtain the value (sensory) showed in table 2. Each sensorial parameter evaluated must be analyzed or correlated in an individual way or define clearly a global score for the appearance, flavor, texture, and taste parameters (with the weight of the parameter). Overall acceptability includes aspects of the other parameters. By the other hand, there is no sense to correlate a parameter as viscosity with taste or flavor. The reviewer suggests rewriting the paper considering the proper use of sensorial scores and resubmit.
Other comments:
Line 92 “…nozzle diameter size of 0.8 mm, nozzle height of 5.0 mm” in line 348 ”…nozzle diameter of 1.0 mm, nozzle height of 3.0 mm”, which is the correct?
Line 134 “250 mL of water”.
Line 141 …and heated in a water bath (100 °C, 20 min and 300 rpm). Is the evaporation measured?
Line 232 “…where sensory score (Y) is expressed as the actual value, and the variables are in terms of coded factors.”, what means in terms of coded factors?
Reviewer 3 Report
Dear Authors,
Considering that Foods is a renowned Journal, I recommend the manuscript "Optimization of the formulation and properties of 3D-printedcomplex egg white protein objects" for minor revision. . I think that this manuscript shows methods and results groundbreaking in the field of 3D food printing, mainly in the sensory evaluation and tribological measurement. I have only some points to improve the manuscript:
What did you rely on for this recipe? Why this ingredients? Correction of some words that are together as Line 251. Correction in number + unit, it is necessary to include a space between them. Correction of the item 2.2. (3D printing). The results are very interesting; however, I suggest to add some explanations about the effect of the ingredients in the final properties (viscosity, sensorial, microstructures etc).
Round 2
Reviewer 1 Report
The second version is acceptable. The authors have well addressed the issues raised in the first revision.
Reviewer 2 Report
No comments